# A Prototype of a Lightweight Structural Health Monitoring System Based on Edge Computing

**DOI:** 10.3390/s25185612

**Published:** 2025-09-09

**Authors:** Yinhao Wang, Zhiyi Tang, Guangcai Qian, Wei Xu, Xiaomin Huang, Hao Fang

**Affiliations:** 1Department of Civil Engineering, Faculty of Civil Engineering and Mechanics, Kunming University of Science and Technology, Kunming 650500, China; wangyinhao@stu.kust.edu.cn (Y.W.); tang@kust.edu.cn (Z.T.); qianguangcai@stu.kust.edu.cn (G.Q.); xuwei@kust.edu.cn (W.X.); 2Intelligent Infrastructure Operation and Maintenance Technology Innovation Team, Faculty of Civil Engineering and Mechanics, Kunming University of Science and Technology, Kunming 650500, China; 3Yunnan Transportation Science Research Institute Co., Ltd., Kunming 650200, China

**Keywords:** structural health monitoring system, long-span bridge, extreme events detection, deep learning, data fusion

## Abstract

Bridge Structural Health Monitoring (BSHM) is vital for assessing structural integrity and operational safety. Traditional wired systems are limited by high installation costs and complexity, while existing wireless systems still face issues with cost, synchronization, and reliability. Moreover, cloud-based methods for extreme event detection struggle to meet real-time and bandwidth constraints in edge environments. To address these challenges, this study proposes a lightweight wireless BSHM system based on edge computing, enabling local data acquisition and real-time intelligent detection of extreme events. The system consists of wireless sensor nodes for front-end acceleration data collection and an intelligent hub for data storage, visualization, and earthquake recognition. Acceleration data are converted into time–frequency images to train a MobileNetV2-based model. With model quantization and Neural Processing Unit (NPU) acceleration, efficient on-device inference is achieved. Experiments on a laboratory steel bridge verify the system’s high acquisition accuracy, precise clock synchronization, and strong anti-interference performance. Compared with inference on a general-purpose ARM CPU running the unquantized model, the quantized model deployed on the NPU achieves a 26× speedup in inference, a 35% reduction in power consumption, and less than 1% accuracy loss. This solution provides a cost-effective, reliable BSHM framework for small-to-medium-sized bridges, offering local intelligence and rapid response with strong potential for real-world applications.

## 1. Introduction

Bridges play a crucial role in modern transportation systems but face complex challenges that threaten their structural integrity. The escalating pace of urbanization and the rapid development of transportation infrastructure have positioned bridges as increasingly vital components of critical transportation networks. However, with extended service lives, bridge materials are susceptible to wear, corrosion, and fatigue cracking [1], which progressively diminish their load-bearing capacity. Under conditions of high traffic density, factors such as overloading and persistent vibrations further accelerate structural aging [2]. Crucially, many age-related structural degradations are intrinsically concealed and not readily discernible through conventional visual inspections, thereby substantially elevating safety risks. Moreover, bridges are vulnerable to sudden structural damage from natural disasters, including earthquakes [3] and ship collisions [4], as well as from human-induced events. These extreme events are characterized by their high unpredictability and often remain undetected or unreported promptly, potentially leading to severe consequences.

Building on the critical challenges faced by aging bridges, the development and application of bridge structural health monitoring (BSHM) technologies have become essential for ensuring safety and durability. In response to the escalating demands of bridge maintenance and management, BSHM technologies have become indispensable. Traditional BSHM systems typically involve the installation of wired sensors at critical structural locations coupled with centralized data acquisition units. These systems are designed for continuous monitoring of key parameters such as stress, vibration, environmental conditions, and displacement [5,6]. Nevertheless, their implementation is frequently hampered by high wiring costs and complex installation and maintenance procedures [7], rendering large-scale deployment particularly challenging in resource-constrained areas. Additionally, the inherent inflexibility of wired sensor layouts makes system expansion and subsequent optimization both expensive and technically demanding.

To overcome the limitations of traditional wired BSHM systems, recent technological innovations have fostered the development of more flexible and intelligent monitoring approaches. Recent advancements in the Internet of Things (IoT) and wireless sensing technologies have propelled the evolution of Structural Health Monitoring (SHM) systems towards more lightweight and intelligent designs. Wireless Sensor Networks (WSNs), which involve the distributed deployment of wireless nodes across bridge structures, have proven effective in mitigating the complexities of wiring and high maintenance costs associated with traditional systems. This paradigm shift significantly enhances system scalability and deployment flexibility [8]. Despite these advantages, WSNs still encounter several technical challenges, including the limited battery life of individual nodes [9], relatively high per-node costs [10], inherent instability in data transmission [11], and difficulties in achieving high-precision time synchronization across multiple distributed nodes [12].

To address the challenge of clock synchronization delay, researchers have developed various post-processing strategies, including timestamp estimation and cross-correlation methods [13]. These techniques aim to establish a unified global time reference throughout the network, thereby improving synchronization accuracy and enhancing the reliability of data transmission. However, such post-processing algorithms are typically applied offline after data collection and cannot achieve real-time synchronization; moreover, the underlying synchronization algorithms are often complex. To compensate for data loss during transmission, Tang et al. [14] and Liu et al. [15] pioneered time-series neural network-based data reconstruction methods capable of handling irregular and non-uniform data loss patterns. Furthermore, Rocha et al. [16] explored the development of multi-node vibration monitoring systems using low-cost hardware, providing novel insights into the cost-effective deployment of Wireless Sensor Networks (WSNs).

The efficacy of Bridge Structural Health Monitoring (BSHM) systems hinges not merely on the accuracy and synchronization of data acquisition but critically on their capacity for real-time structural condition assessment and early warning generation. This capability is pivotal for informed bridge operation, maintenance, and safety management. Among these functions, the rapid identification and response to extreme events constitute one of the system’s most vital roles, particularly within the domain of seismic design for bridges. During an earthquake, a BSHM system must be able to record the structural response in real-time, furnishing crucial evidence for evaluating seismic performance. However, relying exclusively on seismic station data is often insufficient to fully capture the actual loading and dynamic response characteristics of a bridge, owing to the inherent time delay between seismic wave propagation and its arrival at the bridge structure [17]. Consequently, the accurate extraction of earthquake response information from large volumes of monitoring data presents a significant technical challenge [18]. The precise identification of earthquake onset time, duration, and the isolation of pure structural response data are of substantial engineering value for optimizing bridge design and enhancing seismic resilience [19].

In response to the growing need for intelligent recognition of extreme events in Structural Health Monitoring (SHM), a multitude of research initiatives have emerged recently. For instance, Chen et al. [20] effectively identified the natural frequencies and modal damping characteristics of the Tsing Ma Bridge during Typhoon Victor by integrating Empirical Mode Decomposition (EMD) with the Hilbert Transform (HT). Similarly, Thanh-Canh et al. [21] utilized the Stochastic Subspace Identification method and Short-Time Fourier Transform (STFT) to investigate the evolution of dynamic characteristics of a bridge subjected to two distinct typhoon events. Within the specific domain of seismic response identification, Karianne et al. [22] introduced a Fast Seismic Triggering (FAST) method grounded in waveform similarity. This method enables rapid detection and localization of earthquake events via a “waveform fingerprint plus similarity threshold” strategy. Collectively, these studies provide both theoretical underpinnings and technical support for the identification of bridge responses under extreme loading conditions, thereby advancing the development of SHM systems toward enhanced intelligence.

More recently, the widespread adoption of computer vision and deep learning technologies has significantly bolstered the responsiveness and early warning capabilities of SHM systems for identifying extreme events [23,24]. For example, Qian et al. [25], Tang et al. [26] and Guo et al. [18] developed automated earthquake event detection pipelines by employing image transformation and deep learning methodologies. Wu et al. [27] applied the YOLO (You Only Look Once) object detection algorithm to achieve automated bolt damage identification. However, these conventional approaches predominantly rely on cloud computing platforms, which demand substantial computational resources and extensive network bandwidth, consequently inflating deployment and operational expenses [28]. While such expenditures may be justifiable for large-scale bridges, they often present considerable economic barriers for small-to-medium-sized bridges.

To circumvent the inherent limitations of cloud-based deep learning approaches concerning computational resources, network bandwidth, and deployment costs, edge computing has emerged as a highly promising paradigm. By decentralizing data processing from remote cloud servers to locations proximate to the data source, edge computing effectively mitigates the pressures on bandwidth, reduces latency, and optimizes resource scheduling that is typically associated with traditional cloud systems. Its seamless integration into Bridge Structural Health Monitoring (BSHM) enables on-device local data analysis and event recognition, thereby markedly enhancing the overall system responsiveness and stability [29].

Recent practical implementations have further validated the effectiveness of edge intelligence in real-world bridge monitoring scenarios. Illustratively, Wang et al. [30] developed a wireless monitoring system integrating 5G communication with edge computing, specifically deployed on the Hong Kong–Zhuhai–Macao Bridge. This system primarily performs preliminary AI-driven sensor fault detection to identify data anomalies caused by sensor failures while uploading only fault-free massive datasets to the cloud platform, thereby significantly reducing data transmission bandwidth. Similarly, Chou et al. [31] combined an AIoT (Artificial Intelligence of Things) architecture with a PWO-Lite-optimized YOLOv7-tiny model and the DeepSORT algorithm to effectively identify concrete degradation. However, due to the performance limitations of the edge computing platform, the inference speed remained relatively low, achieving only 10 frames per second under FP16 half-precision inference. This work nonetheless demonstrates the considerable potential and challenges of deploying edge intelligence in practical bridge monitoring applications.

To advance cost-effective and intelligent Bridge Structural Health Monitoring (BSHM) and directly address persistent challenges in Wireless Sensor Networks (WSNs) as well as the cost and latency limitations of cloud-based Artificial Intelligence (AI), this study presents a novel, fully integrated BSHM system that combines lightweight design principles with an edge computing architecture. The system features ultra-low-power, cost-efficient distributed wireless sensing nodes paired with an intelligent hub equipped with a GNSS module to establish a local NTP calibration server, significantly improving clock synchronization accuracy beyond traditional cloud-based solutions by minimizing physical calibration distance. Data packet handling is optimized through packet merging, raw binary transmission, and a protocol with buffering, acknowledgments, and retransmissions, substantially enhancing wireless transmission reliability and reducing bandwidth and power consumption compared to existing systems. The deployment of a quantized, lightweight MobileNetV2 neural network model on an NPU-enabled edge platform enables real-time structural acceleration data acquisition and immediate earthquake event recognition, achieving markedly faster inference speeds and lower energy usage than previous approaches that relied solely on preprocessing or slower edge inference. Collectively, these hardware-software co-design innovations not only reduce total energy consumption, physical footprint, and deployment costs but also deliver a more robust, responsive, and practically deployable monitoring solution, representing a significant advancement over conventional BSHM frameworks.

The primary contributions of this study are as follows:Innovative Wireless Node Design and High-Precision Synchronization Mechanism: We proposed a lightweight and cost-effective wireless sensor system that integrates a Global Navigation Satellite System (GNSS) module within the intelligent hub to establish a local Network Time Protocol (NTP) server, thereby significantly enhancing clock synchronization accuracy across distributed nodes. Furthermore, the data packet structure was optimized, a buffering mechanism was introduced, and a transmission protocol incorporating acknowledgments and retransmissions was implemented, effectively strengthening anti-interference capability and ensuring data transmission integrity.Comprehensive System Performance Validation: The system’s performance was systematically evaluated through vibration tests on a steel bridge model, dedicated clock synchronization assessments, and large-scale wireless transmission experiments. The results demonstrated excellent performance in data acquisition accuracy, synchronization precision, transmission reliability, and resilience to electromagnetic interference.Lightweight Edge AI Model with Efficient Inference: A lightweight neural network based on MobileNetV2 was developed and quantized to int8 precision to optimize inference performance. Efficient inference was achieved on devices equipped with Neural Processing Units (NPUs). Experimental results using real bridge earthquake datasets showed that the model maintains high recognition accuracy while substantially improving inference speed and reducing power consumption.

The structure of the remainder of this paper is as follows: Section 2 elucidates the overall system architecture, detailing both its hardware and software implementations. Section 3 presents the experimental validations of the system’s performance. Section 4 elaborates on the construction of the lightweight model for earthquake event recognition, including its quantization and deployment on edge devices. Finally, Section 5 provides the study’s conclusions and outlines potential avenues for future research.

## 2. Monitoring System Design

This chapter details the lightweight edge-computing-based SHM system developed to address the challenges previously outlined in the Introduction. Reflecting the functional requirements and the inherently distributed nature of monitoring, the system architecture is composed of two primary components: sensor nodes and an intelligent hub, as depicted in Figure 1. The sensor nodes are strategically deployed at critical locations across the bridge structure. They function as the front-end data acquisition units, responsible for the real-time collection and preliminary processing of high-precision acceleration data. Conversely, the intelligent hub serves as the core of the system. It is typically positioned to ensure optimal wireless coverage and is tasked with aggregating data from all sensor nodes, managing persistent data storage, maintaining high-precision time synchronization, and conducting edge-intelligent analyses, such as earthquake event recognition.

### 2.1. Hardware Design of Sensor Nodes

As the fundamental data acquisition unit within a distributed monitoring system, the hardware design of the sensor node is paramount to the overall system’s reliability and data quality. This study meticulously considered the practical requirements of BSHM, prioritizing a compact, low-power, reliable, and readily installable node design. The designed sensor node integrates several key functional modules, including an accelerometer, a wireless microcontroller, power management circuitry, and essential communication interfaces. All critical components are assembled on a custom-designed Printed Circuit Board (PCB). The wireless microcontroller serves as the central coordinator, orchestrating data acquisition and wireless transmission. A dedicated power management module (or battery-powered circuit board) ensures a stable power supply to the device. The internal structure and the PCB layout of the sensor node are presented in Figure 2.

The system’s core controller, the ESP32-S3 by Espressif Systems, features a dual-core Xtensa LX7 processor and runs FreeRTOS for robust multitasking. One core manages network protocols, while the other executes application logic, enhancing efficiency and responsiveness. Supporting IEEE 802.11 b/g/n Wi-Fi with speeds up to 150 Mbps and 22 dBm transmission power, it ensures reliable medium-to-long-range wireless communication for bridge structural health monitoring.

In selecting the core sensing component, the accelerometer, a careful balance was maintained among performance, power consumption, size, and cost. This study ultimately adopted the ADXL355, a tri-axis MEMS (Micro-Electro-Mechanical Systems) accelerometer manufactured by Analog Devices Inc., Norwood, MA, USA. This sensor features a differential signal path design that effectively reduces noise and drift, and it incorporates a built-in digital filter to further enhance signal quality. A significant advantage of the ADXL355 is its register-based configurability, allowing for the adjustment of performance parameters such as measurement range, output data rate (ODR), and filter cutoff frequency, thereby enabling optimization for specific application scenarios.

For typical bridge SHM, vibration amplitudes are generally within ±1 g, and the fundamental natural frequencies of medium-to-long-span bridges typically fall below 20 Hz, with most higher modes rarely exceeding 50 Hz [32,33,34]. Similar frequency characteristics are also observed in other civil structures [35]. Based on these conditions, the measurement range was set to ±2 g to ensure coverage of dynamic responses without saturation while providing adequate resolution for small-amplitude vibrations. To achieve sufficient data fidelity for modal analysis and spectral estimation, the ODR was configured to 250 Hz, which satisfies the Nyquist criterion for the target frequency band and ensures accurate capture of structural vibration characteristics.

To mitigate the inherent clock accuracy limitations of MEMS sensors—which often exhibit significant errors, such as the ADXL355’s internal clock error of approximately 1668 ppm [36]—and to ensure stable, high-precision sampling rates, this study leverages the high-resolution timer embedded in the ESP32-S3. This timer offers an accuracy better than ±10 ppm [37] and a time resolution of 1 μs as an external synchronization clock source to drive the ADXL355 at the selected output data rate (ODR). The sensor is configured to receive this external synchronization signal and operate in interpolated filter mode, utilizing the ESP32-S3’s high-precision timing pulses to precisely control sampling intervals. This approach substantially enhances the temporal accuracy and stability of data acquisition, a critical factor for achieving high-precision clock synchronization among distributed nodes and ensuring reliable vibration analysis. The key technical specifications of the ADXL355 are summarized in Table 1.

The sensor housing is manufactured from ABS material using 3D printing, resulting in a compact and lightweight design with external dimensions of 75 mm × 50 mm × 45 mm. The bottom of the enclosure incorporates six designated holes for mounting powerful neodymium magnets, which facilitate rapid, secure, and non-invasive installation on steel structures, as depicted in Figure 3. A magnetic snap-fit mechanism connects the top cover to the base, ensuring both structural robustness and environmental protection. This design also allows users to effortlessly monitor the device’s operational status on-site via internal indicator LEDs, thereby simplifying both deployment and maintenance procedures.

### 2.2. Software Design of Sensor Nodes

The sensor node firmware is critical for achieving high-precision data acquisition, preliminary local processing, and reliable wireless transmission. This firmware is developed using Espressif’s IoT development framework (ESP-IDF) and is built upon the FreeRTOS real-time operating system kernel. By leveraging the high-performance dual-core processor of the ESP32-S3 and the multitasking capabilities of FreeRTOS, the system can concurrently manage high-throughput, real-time data acquisition and transmission tasks. This approach offers significant advantages in complex application scenarios compared to traditional bare-metal microcontroller development. To ensure uninterrupted data collection, even amidst unstable network conditions or transmission delays, a substantial data buffer queue is implemented within the ESP32-S3’s pseudo-static RAM (PSRAM). Figure 4 illustrates the main task framework of the sensor node. After system power-up and initialization—which includes Wi-Fi connection establishment, time synchronization, and accelerometer calibration—the core tasks begin parallel execution.

#### 2.2.1. Sensor Data Acquisition Task

This task is responsible for the real-time acquisition of high-precision acceleration data, preliminary signal processing, and data packetization. To ensure a stable sampling rate, the system utilizes a precise PWM (Pulse Width Modulation) square wave, generated by the ESP32-S3’s built-in high-resolution timer, as an external synchronization clock input to the ADXL355 accelerometer. When the ADXL355 captures new data, it signals data readiness by triggering an external hardware interrupt on the ESP32-S3 via its INT pin. The acquisition task then reads the data from the ADXL355’s internal FIFO (First-In, First-Out) buffer, a 96-entry, 21-bit FIFO capable of storing up to 32 sets of three-axis acceleration data.

The raw acceleration data is preprocessed to remove noise. Since the natural frequencies of bridges are generally below 10 Hz, a low-pass filter with an effective cutoff frequency of approximately 62.5 Hz is applied. This filtering is not performed manually but is implemented through the sensor’s built-in two-stage digital filtering structure. The first stage consists of a fixed decimation filter operating at an internal output data rate (ODR) of 4 kHz, with a cutoff frequency of about 1 kHz. The second stage is a variable decimation filter that automatically activates when the ODR is set to 2 kHz or lower [38]. In this study, the sensor is configured with an ODR of 250 Hz, under which the internal filtering chain achieves a cutoff near 62.5 Hz, a group delay of approximately 6.27 ms, and an attenuation of −1.83 dB at ODR/4, as specified by the manufacturer. This filtering strategy effectively suppresses high-frequency noise beyond the structural response range while retaining critical low-frequency vibration components relevant to structural health monitoring.

The filtered data is subsequently packaged into fixed-format data packets, as illustrated in Figure 5. Each packet comprises an 8-byte millisecond-level timestamp, a 1-byte device ID, 4 bytes for battery voltage, and multiple groups of 20-bit three-axis acceleration data. The size of these data packets directly influences both wireless transmission efficiency and the potential risk of FIFO (First-In, First-Out) overflow.

At a 250 Hz sampling rate, experimental results demonstrated that packaging too few data groups (e.g., one set) per packet leads to an excessive wireless transmission frequency (250 Hz), which can overload the communication channel. Conversely, packaging too many data groups (exceeding 28) risks FIFO overflow, given the ADXL355’s 32-set FIFO depth limitation. Comprehensive system tests evaluating power consumption and bandwidth across various packaging scales (see Table 2) identified an optimal solution: packaging 28 data sets per packet. This approach reduces the wireless transmission frequency to approximately 8.93 Hz and maintains FIFO utilization safely below 87.5%. This balance effectively optimizes data acquisition, buffering, and transmission efficiency. Once packaged, the data is queued in PSRAM (Pseudo-static RAM), awaiting wireless transmission.

#### 2.2.2. MQTT (Message Queuing Telemetry Transport) Data Transmission Task

The MQTT Data Transmission Task is responsible for extracting the packaged sensor data from the PSRAM buffer queue and transmitting it to the intelligent hub’s MQTT server. MQTT is a lightweight communication protocol leveraging a publish/subscribe model. Its low overhead and minimal bandwidth requirements make it particularly well-suited for resource-constrained devices and the often-unstable network environments characteristic of IoT (Internet of Things) applications.

This task operates in a continuous loop, constantly checking the queue for data availability and initiating data transmission whenever data is present. The system employs Quality of Service (QoS) level 1, which guarantees “at least once” delivery. This means the sender repeatedly transmits the data packet until an acknowledgment is received from the receiver, ensuring reliable delivery. MQTT QoS offers three distinct levels:QoS 0 (At Most Once): Messages are sent a single time without acknowledgment or retransmission. This offers the fastest transmission but allows for potential message loss.QoS 1 (At Least Once): Guarantees that messages are delivered at least once, with retransmissions occurring until acknowledged. This level may, however, result in duplicate messages.QoS 2 (Exactly Once): Utilizes a four-step handshake mechanism to ensure messages are delivered precisely once, with no loss or duplication. This level incurs the highest overhead.

This system adopts QoS 1 as a balanced approach, providing reliable data transmission while effectively managing network overhead. This makes it particularly suitable for bridge monitoring scenarios, where high data integrity is required even under unstable network conditions.

Furthermore, to enhance on-site operational visibility, the system toggles an LED indicator’s IO level each time a data packet is successfully sent, offering real-time feedback on transmission status.

#### 2.2.3. NTP (Network Time Protocol) Time Calibration Service

Clock synchronization among distributed sensor nodes is fundamental for ensuring that monitoring data can be effectively utilized for collaborative analysis and precise event localization. Recognizing that the internal crystal oscillator frequency of microcontrollers can drift due to temperature fluctuations and other environmental factors, leading to clock inaccuracies [39], this system integrates a periodic Network Time Protocol (NTP) time calibration service. This service connects to a high-precision local NTP server hosted by the intelligent hub (further detailed in Section 2.4.4) and performs synchronization at regular, predefined intervals (e.g., every ten minutes). By leveraging the NTP calibration algorithm, each node can precisely correct its individual clock deviation, thereby maintaining high-precision synchronization with the local NTP server. This approach significantly enhances the temporal reliability and consistency of data acquired across multiple sensor nodes.

### 2.3. Hardware Design of the Intelligent Hub

The intelligent hub, serving as the core unit for data aggregation, storage, and edge processing, was designed to meet the demands of high AI inference performance, scalable storage, extensibility, low power consumption, and stable operation under field conditions. To achieve a balance between performance and cost, key hardware components were carefully selected.

The Orange Pi 5 was chosen as the central processing platform for its optimal combination of edge AI capability and affordability. Equipped with a 64-bit octa-core ARM processor, up to 16 GB RAM, and a built-in 6-TOPS tri-core NPU, it supports real-time neural network inference for structural health monitoring. Compared with platforms such as the Raspberry Pi (lacking an NPU) or more expensive solutions like the NVIDIA Jetson, it offers an efficient and cost-effective alternative [40,41]. A 500 GB M.2 SSD provides sufficient storage for high-frequency, long-term vibration data, while abundant I/O interfaces enable seamless integration with additional sensors and communication modules.

For stable data transmission and remote access in infrastructure-free environments, the system incorporates a Pugongying X4C 4G router, which aggregates data from wireless sensor nodes and connects to the hub via LAN. Using a 4G IoT SIM card, it enables wide-area connectivity with low power consumption, built-in VPN, and NAT traversal support, allowing secure real-time data access, remote firmware updates, and continuous communication between the local sensor network and remote monitoring systems.

Accurate time synchronization is achieved using an ATGM336H GNSS module, which provides reliable positioning and a high-precision clock reference, supporting BDS, GPS, and GLONASS with PPS accuracy better than 200 ns [42] and low power consumption (<25 mA). By connecting the PPS signal to the SBC’s GPIO and configuring a local NTP server, the system ensures precise clock alignment between the hub and sensor nodes, thereby improving data reliability and enabling accurate analysis of transient events such as earthquakes.

Environmental conditions are monitored with an STH31 temperature and humidity sensor connected via the I^2^C interface. Although not directly involved in earthquake detection, its real-time measurements provide auxiliary information for assessing structural behavior, detecting material condition changes, and evaluating environmental influences during analysis.

### 2.4. Software Design of the Intelligent Hub

The software system of the intelligent hub functions as the core for data aggregation, storage, processing, and intelligent analysis. All tasks pertaining to receiving, parsing, and persistently storing data from the sensor nodes, alongside system maintenance services, are managed by software deployed on the Orange Pi 5 single-board computer. Figure 6 visually represents the software architecture of the intelligent hub. The system design meticulously accounts for real-time data processing, accuracy, reliable transmission, and future scalability. This section will provide a detailed introduction to the intelligent hub’s key software services, while the edge computing-based earthquake event recognition function will be addressed in a separate discussion in Section 4.

#### 2.4.1. MQTT Message Service

This system employs EMQX, a cloud-native distributed MQTT broker, as the primary message middleware for receiving data from wireless sensor nodes. EMQX is characterized by its high concurrency, low latency, and elastic scalability. Its brokerless architecture and extensible plugin system facilitate the efficient handling of large-scale sensor data streams, making it exceptionally well-suited for the real-time communication demands inherent in structural health monitoring applications.

To efficiently process the binary data uploaded by the sensor nodes, a rule engine is configured directly on the EMQX server. This rule engine is responsible for unpacking and parsing the binary payloads received on designated MQTT topics. Each incoming data packet, measuring 1785 bits in length, contains a 64-bit acquisition timestamp, an 8-bit node ID, 32 bits for battery voltage, and 28 sets of 20-bit three-axis acceleration data (corresponding to time points t0 to t28). The parsed data are then organized by device ID and acquisition date (e.g., “s1_2024_06_11”) and subsequently packaged into SQL request bodies. These requests are then transmitted via HTTP to the database service for persistent storage, supporting subsequent analysis and visualization.

#### 2.4.2. Database Service

Data persistence is a core function of the intelligent hub, responsible for the efficient storage and management of received and parsed sensor data. This system utilizes TDengine, a time-series database specifically optimized for high-frequency time-series data. It serves as the primary storage engine, meeting the demanding performance requirements of SHM systems for handling both high-frequency vibration and low-frequency environmental data.

To enhance scalability and query efficiency, the system adopts TDengine’s “Super Table” mechanism. This mechanism organizes sensor data with identical schemas but distinct tags under unified management. Two Super Tables are created in the database, corresponding to different sensor data types: the Acc Super Table stores high-frequency vibration data collected by accelerometers, while the Meteorological Super Table stores environmental data gathered by temperature and humidity sensors.

Differentiated storage strategies are implemented based on the characteristics of each data type. Vibration data, sampled at a high rate of 250 Hz, generates approximately 21.6 million records daily. To facilitate partition management and efficient retrieval, the Acc Super Table creates one child table per day, partitioned by date. Each record in this table contains a millisecond-level timestamp (int64) and three-axis acceleration values (float). In contrast, temperature and humidity data are sampled at a much lower rate (1 Hz, resulting in approximately 86,400 records per day). Consequently, only a single child table is maintained under the Meteorological Super Table to centrally store all records, each comprising a second-level timestamp (int64), temperature, and humidity values (float).

Given the limited storage resources of edge devices (the Orange Pi 5 is equipped with a 500 GB SSD), the system incorporates a data retention policy. High-frequency vibration data is automatically purged after 30 days to conserve storage space. Conversely, temperature and humidity data, due to their smaller volume, are retained for six months to support long-term environmental trend analysis. Additionally, a 200 MB memory cache is configured to optimize data access performance. This effectively reduces database I/O load during real-time visualization or seismic event identification tasks, thereby improving the overall system response speed.

#### 2.4.3. Environmental Data Collection Service

The independent temperature and humidity acquisition service is responsible for real-time reading of environmental temperature and humidity data from the STH31 sensor and subsequently storing it within the TDengine database. Upon system startup, the service first initializes connections to both the sensor and the database. Data acquisition is then triggered once every second using a POSIX timer. During each trigger, the service reads data from the sensor, performs a CRC (Cyclic Redundancy Check) check to ensure data integrity, and then inserts the timestamped temperature and humidity data into the database via SQL statements. The entire acquisition process is synchronized and controlled by semaphores and timers, and it includes robust error logging for any data read failures. This service is configured to launch automatically upon system boot via Systemd, guaranteeing continuous and autonomous collection, processing, and storage of environmental data.

#### 2.4.4. Local NTP Service

To ensure high-precision time synchronization of distributed sensor data, the intelligent hub integrates a local Network Time Protocol (NTP) service, as illustrated in Figure 7. This service leverages the precise time reference provided by the GNSS module.

The GNSS module periodically outputs Coordinated Universal Time (UTC) data compliant with the NMEA (National Marine Electronics Association) protocol via a UART (Universal Asynchronous Receiver-Transmitter) interface. Crucially, it also supplies a sub-microsecond-level PPS signal through a GPIO pin as an interrupt. The system utilizes the gpsd service to parse and integrate these time signals, reading NMEA data from /dev/ttyS0 and PPS signals from /dev/pps0. The combination of these inputs significantly enhances overall timestamp accuracy.

To clarify the operational mechanism, Figure 7a illustrates the timing relationship between the 1PPS signal and the NMEA data output. The 1PPS signal marks the exact start of each UTC second. Shortly after, the GNSS module transmits the corresponding NMEA-formatted timestamp via UART. Due to serial communication latency, the NMEA sentence arrives slightly delayed (typically ~50 ms), which is a known and stable offset.

The gpsd [43] daemon correlates the received NMEA data with the preceding 1PPS pulse to reconstruct the precise UTC, effectively compensating for transmission delay. This corrected time is forwarded to chronyd [44] via a shared memory (SHM) interface. Chronyd, as a high-performance local NTP server, continuously adjusts the system clock with rapid convergence and low latency.

Figure 7b presents the system architecture implemented on the Orangepi 5 platform. The GNSS module interfaces with the system via UART and GPIO, feeding both NMEA and PPS signals into gpsd. The processed time data is then shared with chronyd, which disciplines the system clock. This architecture achieves sub-millisecond local time synchronization, supporting precise collaborative data acquisition across multiple sensor nodes.

#### 2.4.5. Data Visualization Service

The system employs Grafana as the primary data visualization interface due to its real-time refresh capability and flexibility. Sensor data—including battery status, acceleration, and temperature-humidity measurements—are presented through configurable dashboards, enabling users to intuitively monitor trends and perform preliminary analyses.

## 3. Laboratory Test and Results

Building upon the system design detailed in Section 2, this chapter presents a comprehensive laboratory validation of the proposed lightweight edge-computing Bridge Structural Health Monitoring system. To thoroughly assess the system’s performance across key aspects—specifically data acquisition accuracy, clock synchronization, and wireless transmission stability—three distinct experiments were conducted: a vibration measurement test, a clock synchronization test, and a wireless transmission stability test.

### 3.1. Vibration Measurement Experiment

#### 3.1.1. Experimental Platform

The vibration measurement experimental platform comprises several key components: a commercial accelerometer and a high-precision dynamic signal analyzer for baseline comparison, the custom-designed sensor node developed in this study, a steel bridge model simulating the dynamic characteristics of an actual bridge structure, and specialized software for data analysis and theoretical comparison.

The commercial accelerometer chosen was the Donghua Testing 1A110E IEPE (Integrated Electronics Piezo-Electric) accelerometer. It features a ±100 g measurement range and a sensitivity of 5 mV/(m·s^2^), coupled with a maximum sampling rate of 7 kHz, providing high-precision vibration data acquisition. The accompanying high-precision dynamic signal analyzer, the Donghua Testing DH5930 [45], offers a wide frequency response range (DC to 50 kHz, flat up to 20 kHz) and a maximum continuous sampling rate of 128 kHz per channel, enabling accurate signal capture from the commercial sensor. Our custom sensor node, detailed in Section 2, was used to collect vibration data from the same measurement points.

The core experimental subject is the steel bridge model (shown in Figure 8), designed as a two-span structure with hinged connections between the deck and the base. To ensure stability, the base is constructed from a 15 mm thick steel plate. The model’s external dimensions are 2000 mm × 400 mm × 520 mm, with the deck plate measuring 1800 mm × 100 mm × 5 mm. The material used is Q235 steel, possessing the following key mechanical properties: yield strength of 235 MPa, tensile strength of 375–460 MPa, elongation of 26%, and impact energy of 27 J. This model effectively simulates the vibration response of a real bridge.

Data analysis software processes the collected time-domain data and conducts frequency-domain analysis (e.g., Fast Fourier Transform (FFT)). Finite element software MIDAS (version 2021) is employed to construct the steel bridge model and calculate theoretical modal parameters, serving as a reference benchmark.

#### 3.1.2. Experimental Method

The experiment was conducted in a laboratory setting. To capture vibration responses under identical excitation conditions, the Donghua Testing 1A110E commercial accelerometer and the custom-designed sensor node developed in this study were mounted side-by-side at adjacent locations on the steel bridge model (as shown in Figure 9). The experimental excitation originated from ambient environmental sources—specifically, low-amplitude random vibrations induced by pedestrian foot traffic in both the laboratory corridor and within the lab itself. This ambient excitation is representative of low-frequency structural vibrations (typically <10 Hz), which are sufficient to excite the fundamental modes of the steel bridge model without requiring external force input. This setup allowed for continuous vibration data acquisition from the steel bridge model for 30 min, enabling the capture of its fundamental frequency and other dynamic characteristics.

Additionally, the MIDAS finite element analysis software was used to perform numerical simulations. Detailed geometric parameters of the steel bridge model and the mechanical properties of Q235 steel were input into the software to calculate the model’s theoretical natural frequencies. These theoretical results served as a reference standard, against which the data accuracy of our custom sensor node was verified during the actual experimental testing.

#### 3.1.3. Experimental Results and Analysis

Figure 10a displays the time-domain waveform of the vibration signal acquired by the custom sensor node from the steel bridge model under natural excitation. To analyze the structure’s natural frequencies, an FFT was applied to this time-domain data. The resulting frequency spectrum, presented in Figure 10b, clearly illustrates multiple prominent frequency peaks.

The frequencies identified by the custom sensor node were compared against both finite element (FE) simulation results and measurements from a commercial sensor, as detailed in Table 3. Analysis of the FFT spectrum confirmed that the custom node successfully identified the first four natural frequencies of the steel bridge model, demonstrating its robust frequency recognition capability. Compared to the FE-derived modal frequencies, the measured results exhibited an approximate 5% deviation. This discrepancy is primarily attributed to the simplified numerical model not accounting for the sensor’s mass, which can influence the dynamic characteristics of small-scale structures.

Critically, the results obtained from our custom node show a high level of agreement with synchronized measurements from the IEPE commercial accelerometer, with frequency discrepancies less than 0.01%. This strong consistency confirms that the custom sensor node developed in this study achieves data acquisition accuracy and performance comparable to commercial-grade sensors for bridge vibration monitoring.

It is important to note that, while sensor mass may influence modal analysis in small laboratory models, its impact becomes negligible in real-world, large-scale bridge structures, where the sensor’s mass is insignificant relative to the total structural mass. Therefore, the custom sensor node offers high reliability and practical applicability for SHM of actual bridges.

### 3.2. Clock Synchronization Experiment

#### 3.2.1. Experimental Method

To evaluate the time synchronization accuracy of the sensor nodes, two sensor nodes were configured to connect to the same NTP server. One node was placed on the deck of the steel bridge model, while the other was positioned directly beneath it. This setup ensured that both nodes experienced identical excitation and could capture similar vibration response waveforms, as shown in Figure 11. An instantaneous excitation was generated by lightly striking the bridge surface with a hammer, which simultaneously triggered both sensor nodes to begin data acquisition.

The primary objective of this experiment was to assess synchronization precision by comparing the waveforms recorded by the two nodes. Cross-correlation analysis or direct comparison of waveform features—such as peaks, troughs, or zero-crossing points—was used to calculate the time offset between the recorded data streams. The measured time difference directly reflects the effectiveness of the synchronization method.

To ensure that the observed time offset primarily stemmed from the NTP synchronization process rather than long-term drift of internal oscillators, the excitation event was performed immediately after the nodes completed their NTP synchronization routine. Each sensor node operated at a sampling rate of 1000 Hz during the test.

To comprehensively assess synchronization performance, multiple NTP server configurations were tested and compared, including:Local GPS-based NTP server: 192.168.1.10 (provided by the edge-computing central node via the GNSS module, as detailed in Section 2.4.4).Public NTP server 1: ntp.aliyun.com.Public NTP server 2: cn.ntp.org.cn.Public NTP server 3: ntp.tencent.com.

#### 3.2.2. Experimental Results and Analysis

Figure 12 illustrates a comparison of the time-domain vibration waveforms captured by the two sensor nodes. These waveforms were acquired after time synchronization using both the local NTP server and a public NTP server (specifically, Alibaba Cloud is presented as an example).

When utilizing the local NTP server, the two sensor nodes initiated data recording almost simultaneously upon detecting the hammer strike, resulting in nearly perfectly overlapping acceleration curves. No significant time offset or phase difference was observed, indicating an extremely small synchronization error. This robustly demonstrates that the local NTP service integrated into the intelligent hub is capable of delivering high-precision time synchronization performance.

In contrast, synchronization employing public cloud NTP servers yielded noticeably degraded performance. Influenced by internet transmission delays and jitter, the data recorded by the two sensor nodes exhibited a clear time discrepancy, in some cases reaching up to 6 milliseconds. This is evident in the acceleration curves as varying degrees of phase shift. Table 4 summarizes the results from multiple synchronization tests using different NTP servers, further confirming that the performance of public NTP services is highly dependent on network conditions and significantly inferior in precision compared to the local NTP service.

The experimental results unequivocally demonstrate that implementing a local NTP (Network Time Protocol) service via the intelligent hub significantly enhances time synchronization performance. This enhancement is achieved by restricting network latency to the local area network (LAN) between the sensor nodes and the hub. This approach substantially reduces both the transmission distance and the path uncertainty of synchronization packets, which are common issues in wide-area networks (WANs). By localizing the time synchronization process, the system effectively minimizes synchronization errors and markedly improves the accuracy and consistency of clock alignment across distributed sensor nodes. This high-precision synchronization is critical for assigning accurate timestamps to transient events, such as earthquakes, and enables reliable multi-node data correlation and analysis. Consequently, the overall reliability and robustness of the monitoring system are substantially enhanced.

### 3.3. Wireless Transmission Stability Experiment

#### 3.3.1. Experimental Method

To evaluate the transmission stability of sensor nodes under wireless network interference, a series of test scenarios were designed to simulate real-world deployment conditions. The core methodology involved maintaining normal data acquisition by the sensor nodes, which wirelessly transmit data to the intelligent hub via a 4G router. Simultaneously, heavy uplink traffic was introduced from other client devices (e.g., iPads) connected to the same router. This simulated channel congestion and interference. Specifically, interference was induced by performing four full-load data uploads per hour, each lasting two minutes, creating periodic disturbances in the wireless communication channel.

In all scenarios, the sensor nodes were configured with a sampling rate of 2000 Hz. Each data packet contained 28 sets of acceleration data, consistent with the optimized packet size defined in Section 2.2.1.

To analyze the impact of data buffering strategies and Quality of Service (QoS) on communication reliability, three distinct experimental conditions were designed:Operating Condition 1 (Small buffer, QoS = 1): The sensor node was configured with a small RAM-based buffer queue capable of holding 10 data packets. Data transmission utilized MQTT with QoS level 1 (“at-least-once” delivery), ensuring message acknowledgment and potential retransmission.Operating Condition 2 (Large buffer, QoS = 0): The sensor node employed a large PSRAM-based buffer queue capable of storing 20,000 data packets. Data transmission used MQTT with QoS level 0 (“fire-and-forget”), which does not guarantee delivery acknowledgment.Operating Condition 3 (Large buffer, QoS = 1): The sensor node used the same large PSRAM-based buffer as in Condition 2 but with MQTT QoS level 1 enabled to provide reliable delivery through acknowledgment and retransmission.

By comparing the packet loss rate, transmission latency, and packet arrival order across these three scenarios, the experiment aimed to assess the impact of wireless network interference on data transmission reliability and validate the system’s robustness and applicability under various configuration conditions.

#### 3.3.2. Experimental Results and Analysis

Figure 13 illustrates the data transmission performance under simulated network interference for Scenarios 2 and 3, both of which utilize large-capacity buffers. The results indicate that, when external interference temporarily reduces wireless transmission speed, data collected by the sensor nodes but not immediately transmitted is temporarily stored in the PSRAM-based buffer queue. By allocating sufficient buffer space, the system effectively prevents data loss due to overflow during periods of network congestion.

Since such interference is typically intermittent or short-lived, all buffered data can be transmitted sequentially to the intelligent hub once network conditions recover. This mechanism ensures both data completeness and transmission reliability, demonstrating the effectiveness of large-buffer configurations in maintaining stable system performance under adverse network conditions.

In contrast, Scenario 1, which utilized a small-capacity buffer, demonstrated significantly poorer performance. As previously noted, when network congestion occurred, the limited internal RAM buffer proved incapable of accommodating unsent data, leading to immediate data loss. This severely compromised the integrity of data transmission and underscored the limitations of insufficient buffering under unstable network conditions.

Table 5 presents a statistical analysis of packet loss across the three experimental scenarios. The results reveal that, although both Scenario 2 and Scenario 3 utilized the same large-capacity buffer, their differing transmission protocols led to significantly varied packet loss rates. In Scenario 2, which employed the QoS = 0 protocol, the sensor node transmitted data packets without waiting for an acknowledgment from the receiver. Consequently, if packets were lost due to network interference, the sender remained unaware and did not attempt retransmission, resulting in unreliable data delivery and a higher risk of packet loss.

Conversely, Scenario 3 utilized the QoS = 1 protocol, where the sensor node awaited an acknowledgment after sending each packet. A packet was deemed successfully transmitted only upon receipt of this confirmation; otherwise, retransmissions were attempted until a predefined retry limit was reached. This acknowledgment mechanism substantially reduced the risk of data loss, ensuring reliable transmission even in the presence of interference.

The experimental results clearly demonstrate that, under wireless network interference conditions, the combination of a large-capacity data buffer (specifically utilizing PSRAM) and a reliable QoS = 1 transmission protocol most effectively ensures the integrity and reliability of sensor data transmission. This integrated approach guarantees the complete and lossless delivery and storage of critical monitoring data to the intelligent central unit, thereby significantly enhancing the system’s robustness.

## 4. Earthquake Event Recognition Based on Edge Computing

Previous chapters have systematically presented the system’s hardware architecture and software design and validated its performance in terms of data acquisition accuracy, sensor node clock synchronization precision, and data transmission security on a laboratory bridge model. This chapter will focus on leveraging edge computing capabilities combined with lightweight deep learning models optimized for edge devices to achieve efficient and accurate online identification of earthquake events from bridge acceleration data. The earthquake event recognition framework is illustrated in Figure 14.

### 4.1. MobileNetV2

The increasing complexity of deep learning models has led to a dramatic surge in their computational resource demands, thereby limiting their deployment on resource-constrained mobile or edge devices. This constraint is particularly critical for Artificial Intelligence (AI) applications such as rapid, low-power, on-site recognition of extreme events in bridge monitoring. Consequently, selecting an appropriate lightweight model is essential.

To address this challenge, Google developed the MobileNet series of lightweight convolutional neural networks, aiming to balance efficiency and accuracy on mobile platforms. As a pioneering effort, MobileNetV1 introduced depthwise separable convolutions, which significantly reduce computational cost [46]. Building upon this foundation, MobileNetV2 further optimized the architecture by incorporating inverted residuals and linear bottleneck layers. These architectural innovations effectively mitigate the feature information loss that can occur due to the excessive reliance on ReLU (Rectified Linear Unit) activations in MobileNetV1, thereby enhancing representational capacity and overall performance while maintaining a lightweight structure [47].

Considering the strict requirements for model size, computational efficiency, and inference speed in edge-based bridge intelligent recognition scenarios, MobileNetV2 was selected as the backbone network for earthquake event identification in this study, owing to its excellent lightweight characteristics and robust recognition performance.

### 4.2. Model Quantization

To further optimize the deployment efficiency and energy consumption of deep learning models on computationally constrained edge devices—particularly platforms equipped with NPU such as the Orange Pi 5—model quantization is of critical importance. Model quantization aims to significantly reduce the model’s memory footprint and computational overhead, as well as accelerate inference, by lowering the numerical precision of model parameters and activations.

The core concept involves converting the model’s floating-point numbers (typically 32-bit floating-point, or FP32) into low-precision numerical representations (such as 8-bit integers, or INT8). This leverages the hardware acceleration capabilities of edge NPUs that are optimized for low-precision computation [48]. By reducing numerical precision, quantization can substantially improve computational efficiency and decrease hardware resource consumption while maintaining relatively high inference accuracy. This makes it especially suitable for the edge computing and real-time recognition scenarios addressed in this study. Common quantization methods include asymmetric quantization and symmetric quantization.

#### 4.2.1. Asymmetric Quantization

Asymmetric Quantization is a quantization method that maps floating-point numbers to integers. It is particularly suitable for processing data with asymmetric distributions. Compared to symmetric quantization, asymmetric quantization introduces a zero-point to better align with the distribution of the original data, thereby reducing quantization error. The formula for quantizing a floating-point number to an integer using asymmetric quantization is:(1)$$p_{quantized}=round(\frac{p}{S})+Z$$
where p is the floating-point value, S is the scale factor, and Z is the zero-point. The result q is the quantized integer value. The operator round() denotes rounding to the nearest integer. For a given bit width B (e.g., B = 8 for INT8 supported by most NPUs), the quantized integer q is typically clipped to the range $\left[-(2^{B-1}),2^{B-1}-1\right]$, depending on whether the quantization is signed or unsigned. This can be expressed as:(2)$$q=clip(round(\frac{p}{S})+Z,q_{\min},q_{\max})$$

The dequantization (i.e., conversion from quantized integer back to floating-point value) is given by:(3)$$p_{dequantized}=(q-Z)\times S$$

Given the clipping range for the original floating-point values $\left[p_{\min},p_{\max}\right]$ and the integer quantization range $\left[q_{\min},q_{\max}\right]$, the scale factor S and zero-point Z are calculated as:(4)$$S=\frac{p_{\max}-p_{\min}}{q_{\max}-q_{\min}}$$(5)$$Z=round(q_{\min}-\frac{p_{\min}}{S})$$

Asymmetric quantization provides a more accurate representation of the true distribution of data, especially for datasets with clearly non-symmetric distributions (e.g., activation values in deep learning models). By using a more flexible quantization range allocation strategy, it improves quantization precision However, the inclusion of a zero-point increases the computational complexity compared to symmetric quantization.

#### 4.2.2. Symmetric Quantization

Symmetric quantization is a simplified quantization method characterized by symmetrically mapping the range of the original floating-point data around zero into the quantized range. Unlike asymmetric quantization, symmetric quantization fixes the zero-point to zero (Z=0), which simplifies the computation and reduces the number of parameters. The quantization formula from floating-point to integer is simplified as:(6)$$p_{quantized}=round(\frac{p}{S})$$

The dequantization formula (i.e., converting fixed-point integers back to floating-point values) is given by:(7)$$p_{dequantized}=q\times S$$

Given the maximum absolute value of the original floating-point data $p_{\max\_abs}$ and the quantized integer range $\left[q_{\min},q_{\max}\right]$ (typically $\left[-2^{B-1},2^{B-1}-1\right]$ or $\left[0,2^{B}-1\right]$), the scale factor S is usually calculated as:(8)$$S=\frac{p_{\max\_abs}}{2^{B-1}-1}$$

The main advantage of symmetric quantization lies in its computational simplicity and ease of hardware implementation, making it suitable for scenarios with high computational efficiency requirements. However, its core assumption—that the data distribution is symmetric around zero—often does not hold in real-world deep learning data. This may result in a loss of quantization precision. For data with asymmetric distributions, symmetric quantization may fail to fully utilize the available quantization range, leading to inaccurate representation of critical data features and negatively affecting model performance. Therefore, the applicability of symmetric quantization depends on the distribution characteristics of specific datasets.

### 4.3. Dataset Selection

Due to the low probability and sudden onset of extreme events, acquiring relevant data through short-term experiments is inherently challenging. As the sensing devices developed in this study have not yet been deployed on actual bridges, historical vibration datasets from a commercial Structural Health Monitoring (SHM) system on a large-span cable-stayed bridge were utilized. The system comprises 70 accelerometer channels installed at critical structural locations, sampling at 50 Hz, thereby providing representative vibration signals under real operational conditions.

Given the rarity of earthquake events, the dataset exhibits a pronounced class imbalance between seismic and normal operational data. To construct a relatively balanced dataset and rigorously evaluate the proposed model, monitoring records from four distinct earthquake events were selected, with one hour of continuous data extracted for each event.

As the selected MobileNetV2 architecture is a two-dimensional convolutional neural network, the original one-dimensional acceleration time series must be converted into image form before model training. Based on an analysis of the SHM data and the bridge’s dynamic characteristics (e.g., a fundamental frequency of approximately 0.2441 Hz, corresponding to a period of about 4.1 s) and with reference to relevant studies [49], key parameters for time–frequency image conversion were determined. To capture sufficient vibration cycles for distinguishing between seismic and normal conditions while ensuring data diversity, a window length of seven vibration cycles was adopted (L = T × 7 ≈ 29 s) and practically set to 30 s for computational convenience. A sliding window strategy with 50% overlap (step size: 15 s) was applied to segment the entire time series. Each segment was transformed into both time-domain waveforms and frequency-domain spectrograms, which were then combined into two-channel time–frequency images [50]. This representation retains complementary temporal and spectral features, providing richer information for model discrimination. As illustrated in Figure 15, seismic samples exhibit sparse long-period waveforms with larger amplitudes in the time domain and energy concentrated in the low-frequency band in the spectral domain—characteristics that sharply contrast with those of normal operational data.

Applying the sliding window and image conversion process to the four seismic events initially produced 67,200 time–frequency images. Subsequent inspection revealed that some samples were affected by sensor malfunctions or transmission errors. After rigorous data cleaning and outlier removal, a final dataset of 42,722 manually verified and accurately labeled images was obtained.

For performance evaluation, stratified random sampling was employed to partition the dataset into training (20%), validation (5%), and test (75%) sets. Class balance between seismic and normal samples was strictly maintained in the training set to prevent bias. This partitioning strategy mitigates the effects of class imbalance, preserves representative data distributions across all subsets, and ensures the scientific rigor and reliability of the evaluation process.

### 4.4. Experimental Platform Setup and Efficient Inference Based on NPU Deployment

#### 4.4.1. Experimental Platform

The experimental hardware platform consists of a PC used for model training and quantization and an Orange Pi 5 development board for model inference. The PC is equipped with an Intel Core i7-12700F processor, 64 GB of RAM, and an NVIDIA RTX 3060 GPU with 12 GB of video memory, providing robust support for high-performance deep learning model development. The Orange Pi 5 development board features the Rockchip RK3588s chip, which integrates a tri-core NPU delivering a total computing power of 6 TOPS and is paired with 16 GB of RAM. This board serves as the core hardware for edge inference of quantized models.

On the software side, model training is conducted using the PyTorch (version 2.0.1) framework. Model quantization and conversion to the RK3588s NPU-compatible format are performed using Rockchip’s RKNN Toolkit 2, which supports quantizing PyTorch models and converting them into an optimized inference format tailored for the RK3588s NPU [51].

#### 4.4.2. Efficient Inference Based on NPU Deployment

After completing the platform setup, the seismic event recognition model was trained on a PC host using the PyTorch framework. The training utilized the processed acceleration dataset described in Section 4.3, with MobileNetV2 as the backbone model tasked with seismic event classification. Training parameters included a batch size of 4 and 20 epochs, alongside data augmentation and normalization strategies (e.g., random flips and mean-variance normalization applied to image-formatted data) to enhance model generalization. To prevent overfitting, partial freezing of the feature extraction layers was employed during training, optimizing only the classification head parameters. The Adam optimizer [52] was used to accelerate convergence, and the cross-entropy loss function guided optimization. The training loop comprised forward propagation, loss calculation, backpropagation, and weight updates, with real-time validation of recognition accuracy and checkpointing of the best-performing model weights.

Upon training completion, to enable efficient execution on the RK3588s NPU, the trained model underwent quantization (e.g., INT8 quantization) and format conversion. The quantization process utilized the Minimum Mean Square Error (MMSE) algorithm to minimize quantization-induced errors and enhance accuracy. The model input was adapted to the NPU’s required dimensions and format (e.g., for 224 × 224 image input, shaped as [1, 3, 224, 224] with appropriate normalization), and a calibration dataset was provided to determine quantization parameters. After quantization, the model was converted to the RKNN format.

During edge inference deployment, the quantized RKNN model was loaded onto the Orange Pi 5’s NPU. The application initialized the RKNN model context and received data streams from sensor nodes, which were preprocessed and converted into the model input format (such as time–frequency images). These inputs were fed to the NPU for accelerated inference, and inference results—e.g., probabilities corresponding to different seismic event classes—were obtained via the RKNN API. By comparing inference outputs with ground truth labels, the recognition accuracy on the edge device was evaluated.

#### 4.4.3. Inference Performance Evaluation

To comprehensively evaluate the impact of different deployment environments on model performance, inference tests of the earthquake event recognition model were conducted across multiple hardware and software configurations. Figure 16 presents the confusion matrices for the original Float32 model and its quantized versions (Float16 and Int8). Table 6 presents a comparison of average inference time and power consumption under five typical deployment scenarios. Specifically, when using NPU inference, the overall utilization of the three-core NPU is approximately between 35% and 45%. The development board utilizes LPDDR4/4x memory with a working frequency of 2112 MHz for its memory bandwidth.

In terms of recognition accuracy, the original float32 model running on the PC platform and the development board’s CPU (Conditions 1–3) all achieved an accuracy of 91.69%. The quantized models exhibited slight accuracy degradation: the float16 model (Condition 4) and the int8 model (Condition 5) showed decreases of 0.13% and 0.60%, respectively, yet still maintained high recognition performance suitable for practical applications.

Regarding inference speed, Condition 1 utilizing the PC GPU (RTX 3060) achieved the fastest average inference time of 2.01 ms, benefiting from its powerful parallel computing capabilities but consuming as much as 110 W of power. The PC CPU (Condition 2) had a longer inference time of 16.74 ms with a power consumption of 90 W. The ARM CPU on the Orange Pi 5 development board (Condition 3) showed a significantly increased inference time of 112.66 ms; despite the slower speed, its power consumption was only 10 W, reflecting the low energy characteristics of embedded platforms.

Deployment of quantized models on the Orange Pi 5 NPU yielded substantial improvements in inference performance. The Float16 quantized model (Condition 4) achieved an average inference time of 6.03 ms, while the Int8 quantized model (Condition 5) further reduced this to 4.23 ms—a 29.85% decrease—accompanied by only a 0.48% drop in accuracy. Power consumption also decreased by 7%. Compared with the unquantized Float32 model running on the development board’s CPU (Condition 3), NPU inference achieved over a 20-fold speedup, with power consumption reduced to 7 W and 6.5 W, respectively. These results highlight the pronounced advantages of deploying quantized models on NPUs for edge devices, markedly enhancing inference speed and reducing energy consumption, thereby significantly improving overall system energy efficiency.

In summary, deploying the quantized earthquake event recognition model on the Orange Pi 5’s NPU—leveraging the NPU’s hardware architecture optimized for neural network computations, parallel processing capabilities, and native support for low-precision arithmetic—achieves substantial improvements in inference speed and energy efficiency with only minimal accuracy loss. This outcome provides a robust technical foundation for realizing localized, real-time, and efficient intelligent recognition of bridge seismic events.

## 5. Conclusions

This study proposes and validates a lightweight wireless bridge structural health monitoring system based on edge computing, addressing key challenges in traditional wired systems—such as complex wiring and high maintenance costs—as well as issues in wireless systems including high node cost, insufficient synchronization and reliability, and the high latency and expense associated with cloud-based intelligent analysis for extreme event detection using computer vision. The system comprises low-cost, low-power sensor nodes and an intelligent hub integrated with an NPU. By leveraging a GNSS module to establish a local NTP server, millisecond-level high-precision clock synchronization among nodes is achieved without the need for post-processing. An optimized wireless data transmission protocol ensures efficient and secure data transfer. Combined with MobileNet model quantization techniques, the system enables efficient earthquake event recognition on edge devices.

Experimental results comprehensively validate the system’s performance: the custom-designed sensor nodes achieve data acquisition accuracy comparable to commercial devices; node synchronization reaches millisecond precision; and, under simulated wireless interference, the introduction of large buffers and MQTT QoS = 1 protocol guarantees high reliability in wireless transmission. The core edge intelligence, based on a quantized MobileNetV2 model, attains over 20-fold acceleration in inference speed on the NPU, with significantly reduced power consumption and less than 1% accuracy degradation, fully demonstrating the energy efficiency advantages of quantized models on edge devices.

In summary, the developed system offers an economical, reliable SHM solution with local intelligent perception and rapid extreme event response capabilities for small-to-medium-sized bridges, holding significant engineering application value and promising potential for widespread deployment.

## Figures and Tables

**Figure 1 sensors-25-05612-f001:**
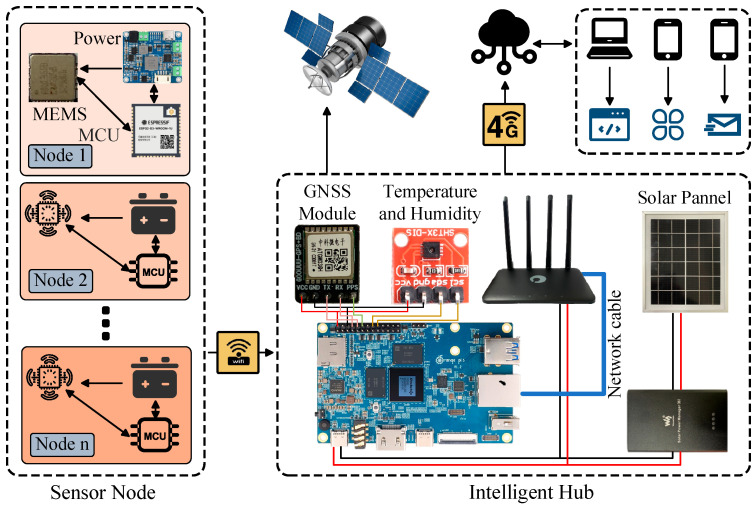
An overview of the system’s operation.

**Figure 2 sensors-25-05612-f002:**
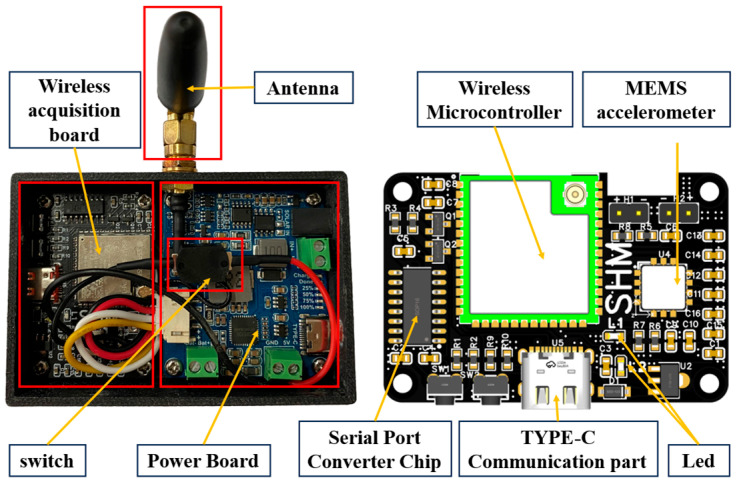
Schematic of the sensor node internals.

**Figure 3 sensors-25-05612-f003:**
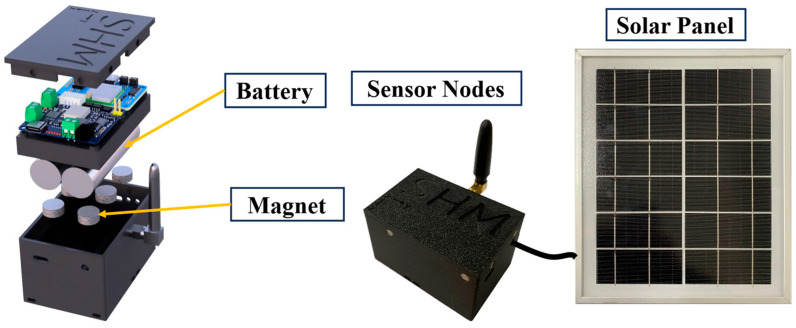
Internal Architecture and Physical Image of the Sensor Node.

**Figure 4 sensors-25-05612-f004:**
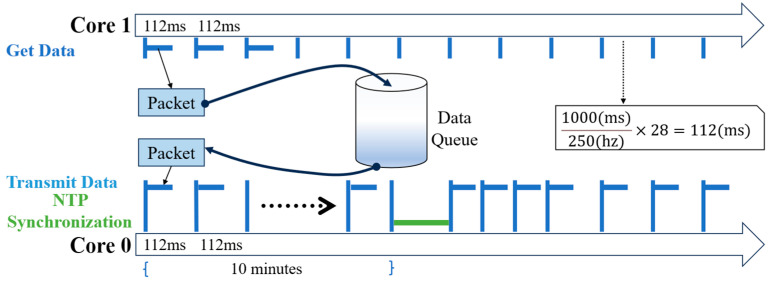
Multitasking operation flow of the sensor node.

**Figure 5 sensors-25-05612-f005:**
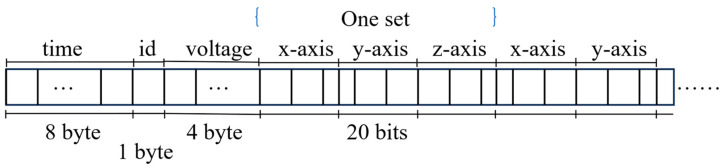
Schematic diagram of the packet structure.

**Figure 6 sensors-25-05612-f006:**
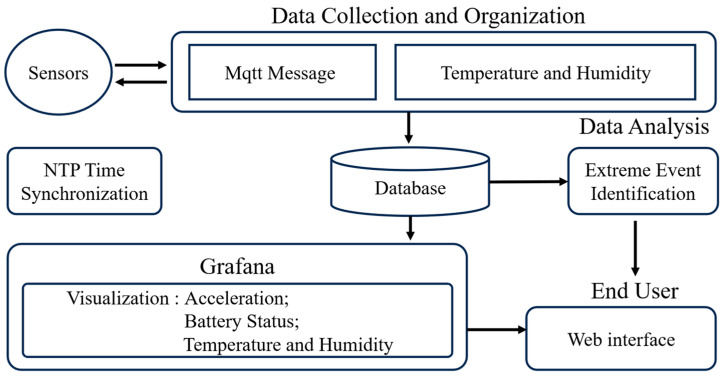
Software framework of the intelligent hub.

**Figure 7 sensors-25-05612-f007:**
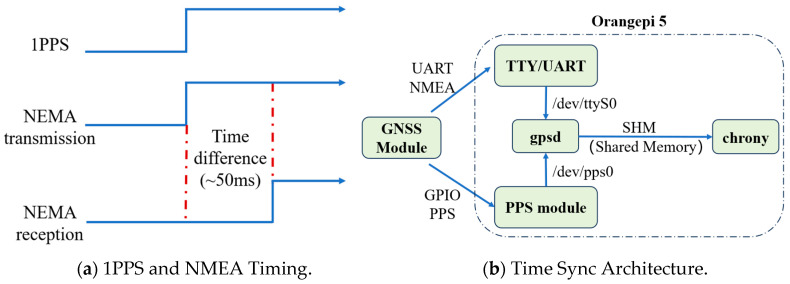
Clock Synchronization Framework.

**Figure 8 sensors-25-05612-f008:**
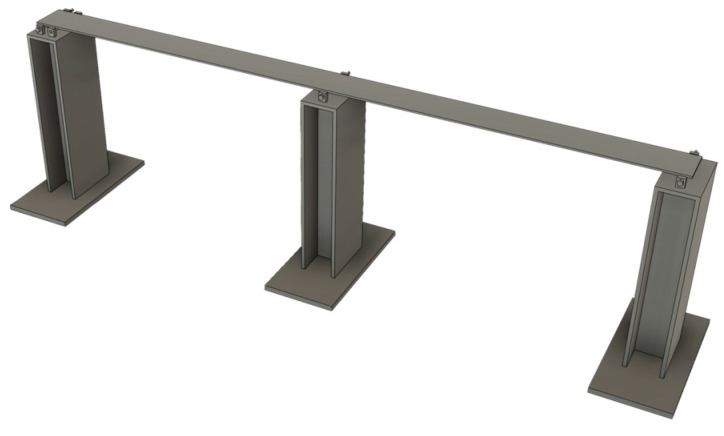
Steel Bridge Model.

**Figure 9 sensors-25-05612-f009:**
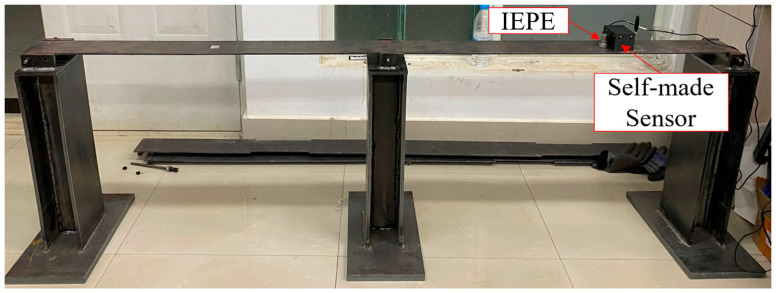
Deployment diagram of IEPE sensor and self-made sensor.

**Figure 10 sensors-25-05612-f010:**
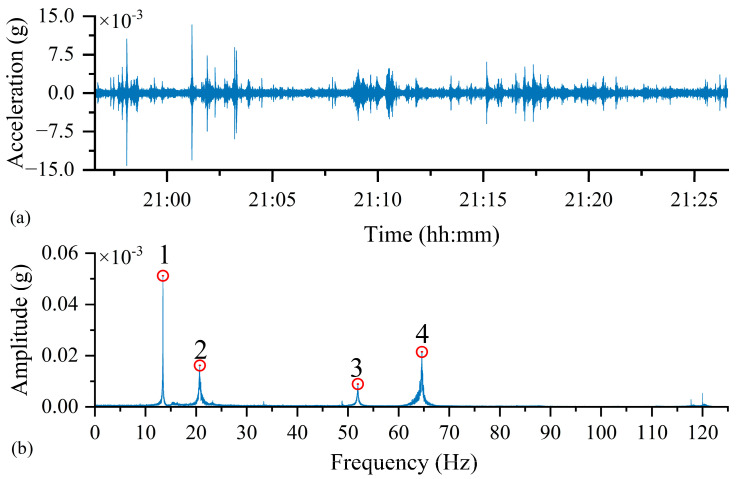
Measurement Results of the Developed Sensor under Natural Excitation: (**a**) Time-Domain Response, (**b**) Frequency-Domain Response.

**Figure 11 sensors-25-05612-f011:**
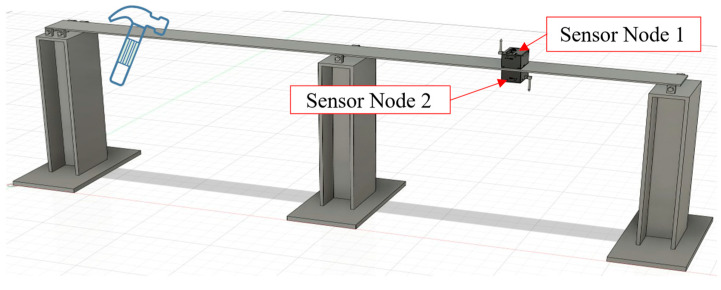
Deployment diagram of two sensor nodes.

**Figure 12 sensors-25-05612-f012:**
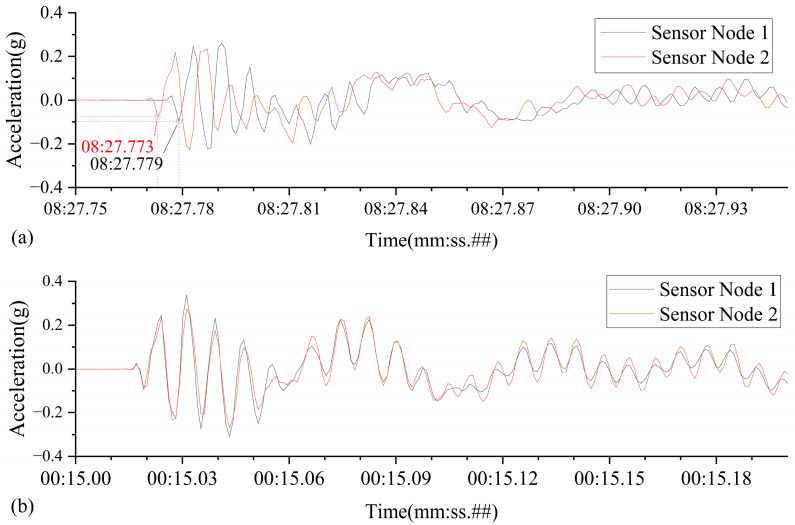
Clock synchronization effect between two sensor nodes: (**a**) using the public server: ntp.aliyun.com; (**b**) using a local NTP server.

**Figure 13 sensors-25-05612-f013:**
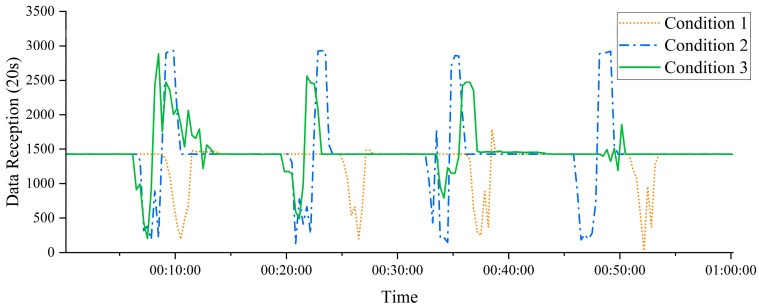
Data Transmission of Sensor Nodes Under Three Operating Conditions.

**Figure 14 sensors-25-05612-f014:**
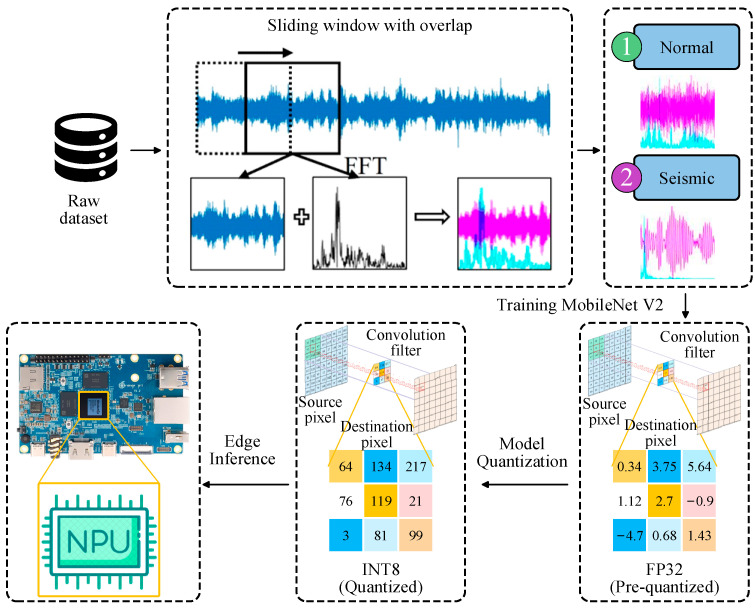
Seismic Event Recognition Framework Diagram.

**Figure 15 sensors-25-05612-f015:**
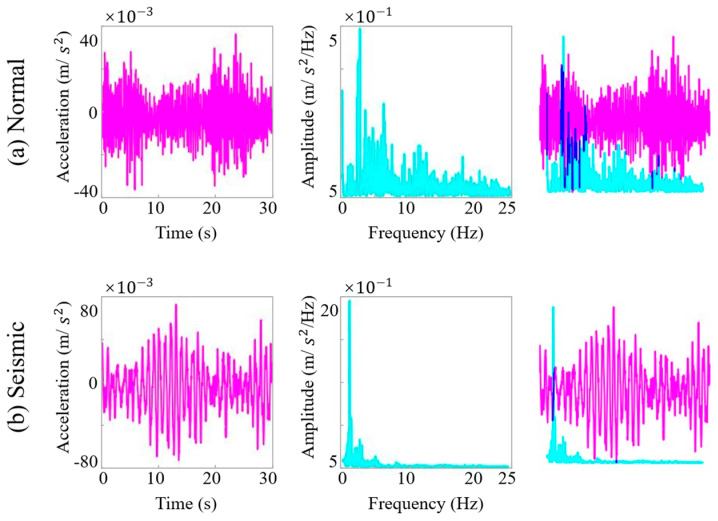
Visual representation of normal and seismic signals in both time and frequency domains.

**Figure 16 sensors-25-05612-f016:**
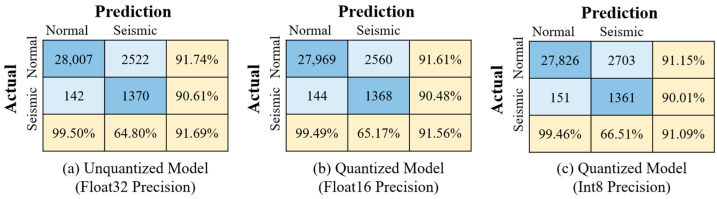
Confusion matrices of the model under different quantization precisions.

**Table 1 sensors-25-05612-t001:** Key parameters of the ADXL355 version [38].

Attribute	Testing Conditions/Remarks	Average Value	Measurement Unit
Sensitivity	±2 g (X, Y, and Z)	256,000	LSB/g
Noise (spectral density)	±2 g (X, Y, and Z)	22.5	μg/√Hz
Sensitivity change due to temperature	−40 °C to +125 °C	±0.01	%/°C
Nonlinearity	±2 g	0.1	%
Zero offset	±2 g (X, Y, and Z)	25	mg

**Table 2 sensors-25-05612-t002:** Power consumption and bandwidth utilization of sensor nodes at a 250 Hz sampling rate for different packet sizes.

Packet Size (Bytes)	2	10	20	28
Average Power Consumption over 5 min (mW)	673.3176	593.1691	584.2711	583.1520
Average RX/TX Rate (kbit/s)	57.73/87.32	12.49/32.34	6.39/24.90	4.74/22.72

**Table 3 sensors-25-05612-t003:** Comparison of modal characteristics measured by different methods.

Mode Numbers	1	2	3	4
Finite Element Analysis (Hz)	13.334	20.829	53.322	67.467
IEPE Sensor Measurement (Hz)	13.130	20.252	51.886	63.765
Proposed Sensor Measurement (Hz)	13.129	20.252	51.885	63.764
Deviation from FEA (%)	−1.537	−2.770	−2.695	−5.489
Deviation from IEPE Sensor (%)	−0.008	0.000	−0.002	−0.002

**Table 4 sensors-25-05612-t004:** Clock deviation between two sensor nodes using different NTP servers.

Configuration	Ping(ms)	Test 1 (ms)	Test 2 (ms)	Test 3 (ms)	Test 4 (ms)
Local GNSS-based NTP server	<1	0	0	1	0
Public Server 1	59	3	1	7	4
Public Server 2	29	1	3	11	4
Public Server 3	29	1	4	2	1

**Table 5 sensors-25-05612-t005:** Packet Loss of Sensor Nodes Under Three Operating Conditions.

Operating Condition	Expected Packets	Received Packets	Lost Packets	Packet Loss Rate (%)
1	257,142	250,062	7080	2.75
2	257,142	235,828	21,314	8.29
3	257,142	257,142	0	0

**Table 6 sensors-25-05612-t006:** Comparison of Inference Performance across Different Hardware Platforms.

Test Condition	Deployment Mode	Accuracy (%)	Average Inference Time per Image (ms)	Power Consumption During Inference (W)
1	GPU inference on desktop (RTX 3060, original float32 model)	91.69	2.01	110
2	CPU inference on desktop (Intel i7-12700F, original float32 model)	91.69	16.74	90
3	CPU inference on Linux-based edge device (original float32 model)	91.69	112.66	10
4	NPU inference on Linux-based edge device (quantized float16 model)	91.56	6.03	7
5	NPU inference on Linux-based edge device (quantized int8 model)	91.09	4.23	6.5

## Data Availability

The data used to support the findings of this study are available from the corresponding author upon request.

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
