# Peer review of "A Prototype of a Lightweight Structural Health Monitoring System Based on Edge Computing"

_sensors, 2025, doi:10.3390/s25185612_

Round 1
Reviewer 1 Report
Comments and Suggestions for Authors
This article aims to investigate a wireless bridge structural health monitoring (SHM) system and its supporting hardware components. The authors developed an in-house remote sensing system and validated it in a laboratory environment. Although they briefly mention potential applications to actual bridges, the manuscript lacks a clear rationale connecting the developed instrumentation with real-world SHM applications. Additionally, several technical issues remain unresolved. Therefore, I believe this article should be reconsidered after major revision for acceptance in Sensors. Below are my specific comments and questions:
- On page 6, it quotes, “Given that structural health monitoring typically focuses on low-frequency vibrations, the measurement range was set to ±2g. To ensure sufficient data resolution and analytical accuracy, the output data rate was configured to 250 Hz.” The authors should specify the sensor requirements—such as the expected vibration amplitudes and frequency ranges—based on typical bridge SHM conditions, and support this with relevant references.
- Section 2.3 is overly lengthy and primarily consists of descriptions of commercially available components. Please revise this section to be more concise, focusing on the core role or contribution of each part within the context of the intelligent hub.
- Regarding the technical approach: shouldn't the vibration signals be averaged to enhance the signal-to-noise ratio (SNR)? Can individual events be reliably monitored without accounting for potential noise interference?
- In Section 2.4.5, the authors describe Grafana, a commercially available open-source tool, in unnecessary detail. Additionally, Figure 8 merely displays standard Grafana dashboard elements and is therefore redundant. Please remove Figure 8 and shorten the overall content of Section 2.4.5.
- In the laboratory test setup, were the weights of the attached sensors sufficiently negligible so as not to affect the dynamic properties of the system?
- In Table 3, the presented content corresponds to mode numbers, not mode shapes. Please correct the misuse of terminology and clarify the distinction accordingly."
- In Fig. 13, is there a specific reason why the magnitudes of the sensor data differ between (a) and (b)?
- Although Section 2.4.4 discusses compensating the system clock, the underlying mechanism for achieving high-precision synchronization remains unclear. The authors should clarify the operational principles of the time synchronization process rather than simply listing hardware specifications and software tools.
- The three operating conditions are not clearly defined in the manuscript."
- Include the labels in the x- and the y-axes in Figure 17.
- The authors claim to have developed 'a lightweight wireless bridge structural health monitoring (SHM) system.' However, the basis for calling the system 'lightweight' is unclear. Please clarify the comparative foundation—specifically, how this system differs from or improves upon existing SHM systems in terms of weight, complexity, or deployment effort.
- Check the Section numbers once again (e.g., 3.4.2 à 4.4.2).
- The objective of Section 3.3 is unclear, particularly in relation to the contribution of the proposed system. Since the authors did not conduct actual experiments using their own SHM system, the relevance of this dataset and its role in validating the system should be explicitly clarified. Please justify the inclusion of this section and clearly explain how it supports the goals of the study.
Author Response
Please review the response in the attachment.

Reviewer 2 Report
Comments and Suggestions for Authors
In response to the significant demand for long-term monitoring of medium and small-span bridges, this paper proposes a lightweight design method for monitoring systems and verifies it through experiments. The paper plays an important role in promoting the development of bridge health monitoring. The following comments are for the guidance of authors:
- The full name should be given for the first appearance of an abbreviation, such as NPU in the abstract.
- The literature review in the introduction is rather disorganized, making it impossible to clearly obtain the progress and shortcomings of the current research.
3. The standardization of the article needs to be further enhanced, including the clarity of the pictures.
Author Response

(The authors gave the same response as above.)

Reviewer 3 Report
Comments and Suggestions for Authors
1. A low-pass filter with a cutoff frequency of 62.5 Hz has been applied. Could you please specify the type of filter used?
2. The experimental excitation was derived from ambient environmental sources. Could you clarify this statement? Additionally, a quantitative measure of excitation intensity should be provided to ensure experimental reproducibility.
3. In Section 4.4.3, could you please provide the NPU utilization rate and DDR bandwidth data?
4. In Figure 1, the term "intelligent hub" is used; however, in the abstract, "edge computing unit" is mentioned, and in Section 2.3, "central hub" is used. Are these terms referring to the same component? Please clarify.
Author Response

(The authors gave the same response as above.)

Round 2
Reviewer 1 Report
Comments and Suggestions for Authors
1. “Figure 16. Examples of Normal and Seismic Patterns” needs to be improved by including x- and y-axis labels. The authors did not address my previous comment.
2. I am not sure why Section 4 (Earthquake Event Recognition Based on Edge Computing) is included in this paper. The authors appear to be introducing a conceptual design without presenting meaningful results. Please remove Section 4 if you do not include the AI recognition results—following the procedure in Fig. 14—from the lab test results.
3. Much of the content is overly lengthy relative to its scientific significance. I suggest reducing sections such as instrument catalogs.
4. As a result, the paper does not consider an actual bridge system, even though the content appears to address bridge testing.
1) Revise the paper title to clearly indicate that it presents a conceptual study.
2) Remove all content related to actual bridge systems unless test results from real bridges are provided.
3) Include AI-based recognition results derived from laboratory test data.
Author Response

(The authors gave the same response as above.)

Reviewer 3 Report
Comments and Suggestions for Authors
1.The introduction (Pages 1–3) does not clearly articulate how the proposed system improves upon existing SHM systems beyond general claims of cost and latency reduction.
2.The quantization process for MobileNetV2is described, but the impact of quantization on model performance is not fully quantified.
3.While the system’s integration of edge computing and lightweight AI is novel, the manuscript does not sufficiently differentiate its contributions from existing edge-based SHM systems.
4.The discussion of practical deployment is limited, and the use of historical datasets raises questions about the system’s real-world applicability.
5.The format of Reference 43 is wrong.
Author Response

(The authors gave the same response as above.)

Round 3
Reviewer 3 Report
Comments and Suggestions for Authors
Some minor revisions:
1.In the abstract:"The edge-deployed model achieves a 26x increase in inference speed and a 35% reduction in power consumption, with accuracy loss under 1%.",A "26x increase" and "35% reduction" relative to what baseline? The reader must refer to Table 6 in the main text to find the answer (NPU inference vs. ARM CPU inference).
2. The title uses "A Prototype of a Lightweight...". While accurate, the word "Prototype" may seem slightly conservative given the comprehensive lab validation and excellent performance. Consider a more confident title that reflects the work's completeness and maturity.
3. References: the references are listed only 5 from 2024-2025, maybe following references can be taken into sonsideration:
(1)Qiu Y, Zheng B, Jiang B, Jiang S, Zou C. (2025). Effect of Non-Structural Components on Over-Track Building Vibrations Induced by Train Operations on Concrete Floor. International Journal of Structural Stability and Dynamics, 2650180.
(2)Jie Wu, Beilin Han, Yihang Zhang, Chuyue Huang, Shengqiang Qiu, Wang Feng, Zhiwei Liu, and Chao Zou. 2025. "Enhancing Bolt Object Detection via AIGC-Driven Data Augmentation for Automated Construction Inspection" Buildings 15, no. 5: 819. https://doi.org/10.3390/buildings15050819.
Author Response
Please review the feedback in the attachment.
